## [Peer Review File · Development (Cambridge, England)]

Toll receptors mediate tissue intrinsic surveillance against aberrant cells by detecting cell fate aberrations

Anna Frey, Laurin Ernst, Friedericke Fischer, Lale Alpar, Yohanns Bellaiche and Anne-Kathrin Classen

DOI: 10.1242/dev.205006

Editor: Thomas Lecuit

Review timeline

Original submission:	4 June 2025
Editorial decision:	21 July 2025
First revision received:	12 November 2025
Editorial decision:	16 December 2025
Second revision received:	12 January 2026
Accepted:	28 January 2026

Original submission

First decision letter

MS ID#: dev.205006

MS TITLE: Toll receptors mediate tissue intrinsic surveillance against aberrant cells by detecting cell fate aberrations

AUTHORS: Anna Frey, Laurin Ernst, Friedericke Fischer, Lale Alpar, Yohanns Bellaiche and Anne-Kathrin Classen

Dear Anne-Kathrin,

I have now received all the referees reports on the above manuscript, and have reached a decision. The referees' comments are appended below, or you can access them online: please go to .

The overall evaluation is positive and we would like to publish a revised manuscript in Development, provided that the referees' comments can be satisfactorily addressed. Please attend to all of the reviewers' comments in your revised manuscript and detail them in your point-by-point response. If you do not agree with any of their criticisms or suggestions explain clearly why this is so. If it would be helpful, you are welcome to contact us to discuss your revision in greater detail. Please send us a point-by-point response indicating your plans for addressing the referees' comments, and we will look over this and provide further guidance.

Reviewer 1

Advance summary and potential significance to field

The authors investigate the role of the Toll family of receptors in a cell-cell recognition system they call "interface surveillance", where mosaic expression of specific signalling pathway components or cell surface receptors induce i) an actomyosin enrichment at the clone borders ii) an activation of the JNK pathway and iii) apoptosis. They use the *Drosophila* wing imaginal disc to make mosaics of Toll receptors over- and under-expression, asking whether this induces interface

surveillance. Their testing is very comprehensive and nicely exploit the natural patterns of Toll receptor expression to ask what triggers interface surveillance. They convincingly demonstrate that it is steep differences in Toll receptors expression, in particular the long Toll class of receptors, which trigger this response. Interestingly, this response is independent of downstream activation of NF- κ B and rather requires the extracellular domains of Toll receptors, consistent with other studies. In a second half of the manuscript, they make mosaics where they activate a diversity of signalling pathways and show that these systematically change the expression of Toll receptors. Overall, this is a well written manuscript with a comprehensive test of the role of Toll receptors in "interface surveillance", which strengthens the experimental evidence for this concept. I have only some minor comments:

Comments for the author

- 1) Data presentation:
 - The authors might want to clarify for a broader audience at the beginning of the results that Toll-GFP etc lines are tagged proteins. Some information in the material and methods of where the tag lies (NH2 or COOH ends?) might be useful.
 - Legend of Figure 1: need to give more information about how apical (E-cadherin) and lateral (actin) optical sections are acquired: are these maximum projections and where along the apico-basal axis are these collected?
 - scale bars are missing in Figure S1.
- 2) The levels of apoptosis seem to vary quite a bit between different cases of interface surveillance. Also, sometimes apoptosis seems to be more prevalent in the wildtype tissue and in other cases, within the clones. More generally, how is cell elimination via apoptosis linked to interface surveillance? It would be good to have a little more discussion about this aspect.
- 3) Similarly, activation of the JNK pathway seems to differ in spatial localisation between experiments with different Tolls. For example, in Toll6 overexpression, TRE-RFP is at the clonal borders on the wildtype side (Fig. 2D'), while in Toll8 overexpression, TRE-RFP response is more diffuse (Fig. 3F). Could the authors comment on this?
- 4) The authors perform a double RNAi of both Toll-2 and Toll-7 and show that this induces JNK activity (Fig. L). What about actin enrichment?

Reviewer 2

Advance summary and potential significance to field

In this manuscript, the authors characterize the function of long Toll receptors in interface surveillance, a process protecting from the emergence of cell mutations affecting cell-fate patterning. Toll8 was identified in a genetic screen as a potential mediator of surface surveillance. Since Toll receptors have been identified as regulators of myosin polarity during germ band extension, they reasoned they could modulate actomyosin contractility and adhesion in interface surveillance.

They first analyzed the expression pattern of the Toll family using enhancer trap fluorescent lines. They found specific patterns of Toll-2, Toll-7 and Toll-8, and a ubiquitous expression of Toll-1 in the wing disc. They tested the impact of overexpression of Toll-1, Toll-3 and Toll-4 in clones, but found no sign of a role in interface surveillance such as clones smoothening, JNK activation and apoptosis at the interface. The expression of Toll-6 however (not expressed in the wing disc) induced the formation of a smooth interface with JNK activation and apoptosis at the border. For Toll-8, they found that overexpression induces interface surveillance specifically in region where Toll-8 is not expressed, while Toll-8 RNAi induces it specifically in Toll-8 expression domain, indicating that a discrepancy in Toll-8 level is required. They further tested Toll-2 and Toll-7 for their role in interface surveillance. They found that expression of each of these receptors induces interface surveillance markers, in particular in zone of low endogenous expression. However, their successive inactivation has no effect, while inactivation of both drive interface surveillance in their expression domain, suggesting a redundant function.

They found that Myosin, Rok and Ena accumulate at the interface between different levels of Toll-6 and Toll-8, suggesting that myosin regulation is a core function of Toll receptors. Then they tested whether aberrant cell fate, known to induce interface surveillance, modifies the expression of Toll receptors. They found that FkhOE and TkvCA impact the expression of Toll-7, Toll-8 and probably Toll-2 in a region-specific way. Ey however only affects Toll-7 and Toll-2. They further tested the potential implication of Toll receptors at the boundary of tumoral clones and found that Toll 2, 7 and 8 are activated at the border of RasV12 clones. However, their expression pattern is not modified in contexts of cell competition (Myc or Hippo/warts expressing clones). They tried to impair the discrepancy of Toll 2, 7 or 8 expressions at the border of TkvCA and FkhOE clones using RNAi (for Toll-2, 7 and 8) or overexpression (for Toll-8), but failed to detect any defect in interface surveillance at the border of aberrant cell fate clones. Finally, they showed that Toll-6 and Toll-8 function in interface surveillance does not appear to go through the canonical NF- κ B pathway.

Comments for the author

Main concerns:

- * My major concern is that although the observations are interesting, convincing and well presented, everything relies on correlation and there is no functional evidence that Toll receptors are required for the interface surveillance process. This weakens the impact of this work. Inactivating Toll-2 and Toll-7 in a context of Ey clones (ideally in a whole domain as done in Fig S7) should be tried since Toll-8 is not affected in this context (although it may be more complex and necessary to inactivate also Toll-6 if Ey leads to an up-regulation of Toll-6).
- * How important are Toll receptors at the interface compared to Robo and other genes involved in interface surveillance? How redundant? What do we learn about the interface code? This should be discussed.

More specific concerns:

Fig2:

- * The quantification of cell death should be in the principal figure, presented the same way (with interface cells distinguished from the rest).
- * Please explain how circularity is measured.

For Figure 3-4

- * A clear explanation is missing on how the different domains (outside/inside Toll expression domain) are determined for circularity, TRE-RFP and Dcp1 intensity quantification.

Fig3:

- * In C and D, TRE-RFP should be shown alone. Also, a staining with Wg would help determine the different domain of the disc (hinge versus pouch, versus notum). Quantifications are missing for Toll8 RNAi. In I and K, the number of n seems quite low for the notum territory.

Fig4:

- * The characterization of cell death pattern should be shown in the principal figure, together with the related quantifications of cell death at the interface of the clone versus further away. In L, the general view of the disc is missing (with TRE-RFP shown independently), together with a zoom on the pouch region, including actin and Dcp1 staining and the corresponding quantifications in the different domains.

Fig6:

- * All data presented should be quantified, not only Toll-8 for Fkh and Tkv clones.
- * In the text, the author mentioned: "In our analysis, we found that Fkh and TkvCA-expressing clones exhibited aberrant up and downregulation of Toll-7-Venus and Toll-8-YFP, while Ey-expressing clones altered Toll-7-Venus but not Toll-8-YFP expression (Fig 6)." However, the effect on Toll-8 is not the same for Tkv and Fkh. This should be more carefully commented. Tkv induces downregulation of Toll-8, while Fkh induces an up-regulation. In addition, does Fkh induce an upregulation outside Toll-7 expression domain? This is not commented while clearly shown in the figure.

* Why not looking at Toll-6? This is important, independently of the absence of endogenous expression in the wing disc, to explore the idea of the code of interface surveillance.

* Finally, this figure shows that the code is different depending on the type of cell fate perturbation, which is complex, but interesting. The next step would be to test the necessity of these Toll receptor in interface surveillance. Intents have been made, presented in Fig S7 and S8. However, it would be worth focusing on one particular cell fate perturbation context such as Ey clones for example. Indeed, since they do not depend on Toll-8, we could expect less redundancy in these clones between different Toll, and be able to test the role of Toll-2, Toll-7 or both.

Fig S1:

* Please explain A and B. There is too little information to appreciate what these two panels bring.

Fig S6:

* Why using FlyFos-Toll-2 when it does not recapitulate the expression pattern of Toll-2? Why not using Toll-2-GFP used in Fig1 to characterize Toll-2 expression pattern?

* A final recapitulating scheme recapitulating the code of interface surveillance would help.

* Precise the number of discs analyzed for each experiment. This information is missing all along

Others:

* Line 457 "our findings highlight the specificity and sensitivity of Toll receptor-mediated detection of aberrant cells..." The sensitivity of the system is an interesting point to discuss, however, the UAS/GAL4 system induces probably huge discrepancies of Tolls. In addition, the fact that loss of function of only Toll2 or Toll7 does not induce interface surveillance, due to redundancy, suggests that a factor 2 in the discrepancy is not sufficient.

* Fig S8F: it is said that there is no regulation of Dorsal during interface surveillance, however there is an increased signal of Dorsal shown in S8F.

* Line 387: Fig S7M does not exist in this version of the paper. Please correct.

* 402-404: "Future research needs to address if this is reflected in normal imaginal disc development, such that distinct cell fates establish unique cell surface profiles using a combinatorial code of different long Toll receptors, thereby providing information about the spatial position of cells." This is an interesting point, however, how to explain that normal Toll expression borders in the wing do not lead to Interface surveillance? This should be further discussed.

* 433-435: "Whether these non-canonical ligand interactions explain our findings that the extracellular domain of a receptor such as Toll-6 is completely sufficient to induce interface surveillance, remains to be investigated." Please simplify and reformulate.

* Some typos: l.48 "a distinct tissue-intrinsic surveillance system", l.52 "consistently induce", l.71 "all hallmarks", l. 87 "provides" l. 450 "may not to not be" l.458 "represent", l.460 "not just in innate immunity".

* Explicit sc-RNAseq (single cell-RNAseq) an AHS (after heat-shock)

Some overstatements are found all along the manuscript:

The title "Toll receptors mediate tissue intrinsic surveillance against aberrant cells by detecting cell fate aberrations" does not represent the actual data in the present version of the manuscript since no clear proof of the role of Toll receptor in interface surveillance is established.

Line 29: "key mediators" Line 383: "our study demonstrates a novel function of long Toll receptors..." : This is not demonstrated since no loss of function experiments are conclusive on this point.

Line 405: "Our findings emphasize the importance of a spatial code of Toll receptor for the detection of aberrant cell fates..." This is an overstatement. These findings "suggest" the importance of a spatial code.

First revision

Author response to reviewers' comments

To all reviewers:

We thank the reviewers for their thoughtful and constructive feed-back. In response, we have substantially revised the manuscript and added new experimental data and figures, which we hope significantly improved the clarity and conclusion of the manuscript. Specifically, we have added these new data/figures:

Figure	New/changed data
3B-D	Actin staining included in overview panels to visualize morphological landmarks, additional changes to include TRE-RFP images for control and experiment in Fig S3
3L-N	Quantifications of circularity and mean TRE-RFP intensity of Toll-8-RNAi expressing clones
4L,M	Dcp-1 staining in Toll-2 (L) or Toll-7 (M) overexpressing clones
4N,O	Toll-2/-7 redundancy, full pouch display, actin and TRE-RFP staining
4P,Q	Quantification of TRE-RFP intensity in clones co-expressing UAS-Toll-7-RNAi and UAS-Toll-2-RNAi
6	Figure flow changed to a fate-centric view
6A,D	Regulation of CRISPR Toll-2-GFP
6B	Exchanged disc to also visualize upregulation of Toll-7 in pouch
6G,G',H,H',J	New quantifications
6I,I',K,K'	Different presentation of quantifications
8I	Summary/Model Scheme
S1I	Exchanged to Toll-8-GAL4 (FlyFos Toll-2-GFP data removed from manuscript)
S2E	Improved clarity of scheme
S2F,G	Spatially resolved quantifications of Dcp-1 intensity in wing discs with UAS-GFP-expressing control and UAS-Toll-6-expressing clones
S3A-C'	Full disc display of Actin staining (MZP) to visualize morphological landmarks and TRE-RFP staining (single z-slice) of wing disc overviews shown in Fig3B-D
S3D,E	Exchanged WT zoom to represent control disc in Fig 3B, Fig S3A,A'
S3G	Quantifications of mean Dcp-1 intensity in the pouch of wing discs expressing UAS-Toll-8 moved from main Fig 3 to Fig S3,
S3H	Adjusted presentation of Dcp-1 quantification of wing discs expressing UAS-Toll-8 in the notum
S3J	New quantifications of mean Dcp-1 intensity in the pouch of wing discs expressing UAS-Toll-8- RNAi
S4A,B	Spatially resolved quantifications of mean Dcp-1 intensity in wing discs with UAS-Toll-2- expressing or UAS-Toll-7-expressing clones
S4C	UAS-Toll-2-RNAi verification using CRISPR Toll-2-GFP instead of FlyFos Toll-2GFP
S6A	Exchanged FlyFos Toll-2-GFP to regulation of CRISPR Toll-2-GFP by Ey
S6B	New disc to visualize upregulation of Toll-7-Venus by Ey in the pouch
S6D	Quantification of regulation of Toll-7-Venus by Ey in the pouch
S7.1A-D	Replaced all experiments of FlyFos Toll-2-GFP with CRISPR Toll-2-GFP
S7.1G,H	New necessity experiment, double KD of Toll-2 and Toll-7 via RNAi in UAS-Ey-expressing clones
S8F	Exchanged disc to improve visualization of D1 (new experiment with adjusted timepoints to get flat clones to remove optical artefacts)
Notably, we restructured Fig. 6/S6 and added spatial quantifications to more clearly describe how aberrant cell fate perturbations modulate Toll-receptor expression across tissue domains. We substantially edited the Discussion to discuss how interface-proximal and long-range signals as well as potential receptor-specific downstream effects give rise to apoptotic patterns and cell elimination by interface surveillance. In addition, we have made the changes to Materials & Methods to add information for image analysis workflows and fly strains. We standardized reporting of sample sizes (n/N) across figure legends. Finally, while our data provide strong evidence that long Toll receptors are sufficient to drive interface surveillance, we acknowledge current technical limits on testing necessity within	

combinatorial surface-code contexts and report new attempts (e.g., in the Ey background) consistent with this interpretation.

Reviewer 1: SUMMARY OF THE ADVANCE MADE IN THIS PAPER AND ITS POTENTIAL SIGNIFICANCE TO THE FIELD

The authors investigate the role of the Toll family of receptors in a cell-cell recognition system they call "interface surveillance", where mosaic expression of specific signalling pathway components or cell surface receptors induce i) an actomyosin enrichment at the clone borders ii) an activation of the JNK pathway and iii) apoptosis. They use the *Drosophila* wing imaginal disc to make mosaics of Toll receptors over- and under-expression, asking whether this induces interface surveillance. Their testing is very comprehensive and nicely exploit the natural patterns of Toll receptor expression to ask what triggers interface surveillance. They convincingly demonstrate that it is steep differences in Toll receptors expression, in particular the long Toll class of receptors, which trigger this response. Interestingly, this response is independent of downstream activation of NF- κ B and rather requires the extracellular domains of Toll receptors, consistent with other studies. In a second half of the manuscript, they make mosaics where they activate a diversity of signalling pathways and show that these systematically change the expression of Toll receptors. Overall, this is a well written manuscript with a comprehensive test of the role of Toll receptors in "interface surveillance", which strengthens the experimental evidence for this concept. I have only some minor comments:

SUGGESTIONS TO AUTHORS

1) Data presentation:

- The authors might want to clarify for a broader audience at the beginning of the results that Toll-GFP etc lines are tagged proteins. Some information in the material and methods of where the tag lies (NH2 or COOH ends?) might be useful.

Of course. We now mention in the main text that we are evaluating CRISPR-generated tagged fusion proteins. We also included information on C- and N-terminal tag position in Table S1.

- Legend of Figure 1: need to give more information about how apical (E-cadherin) and lateral (actin) optical sections are acquired: are these maximum projections and where along the apico-basal axis are these collected?

We only explained this in the Materials and Methods Section 'Figure Display', where we state: For subcellular localization of Toll-2, Toll-7 and Toll-8, staining of the adherens junction marker E-cadherin was used to project apical surfaces using the local Z projector, and lateral F-actin sections were extracted using the offset function relative to the adherens junction projection. We now refer the reader in figure legends to materials and methods to image display section (we had cut it due to word limits at *Development: Images are derived from LocalZ-projections (see 'Image display' section in Experimental procedures)*).

- scale bars are missing in Figure S1.

Scale bars were presented in Fig. S1C and Fig. S1J intended to apply to the panel series linked by same experimental concept shown at identical magnification. We now state this explicitly in the legend :

Figures S1C-S1I and S1J-S1W are shown at same scale.

2) The levels of apoptosis seem to vary quite a bit between different cases of interface surveillance. Also, sometimes apoptosis seems to be more prevalent in the wildtype tissue and in other cases, within the clones. More generally, how is cell elimination via apoptosis linked to interface surveillance? It would be good to have a little more discussion about this aspect.

- You are completely right.
- The different apoptotic patterns are very likely driven by parallel and locally overlapping signals:
- We previously demonstrated that JNK activation at clonal interfaces sensitizes both mutant and neighboring wild-type cells to apoptosis (Prasad, Illek et al., 2023). the existence of additional and overlapping pro-apoptotic mechanisms acting both near and away from the interface. These may include:
- Mechanical compression at the interface depending on interface curvature, and bulk compression of clones depending on cell growth dynamics (Valon, Matamoro-Vidal et al. 2025)
- Mechanical fluctuations at the clonal interface and differential transduction or propagation of tension within wild type and mutant cell populations (Schoenit, Monfared et al. 2025)
- Long-range apoptotic signals induced by the cell elimination process itself (Perez-Garijo, Fuchs et al. 2013).

We know include boundary resolved quantification of apoptotic patterns for all long Toll-like receptors, were possible: Toll-2 and Toll-8 -expressing clones produce apoptotic patterns consistent with JNK activity concentrated at interfaces. In contrast, Toll-6 and Toll-7 clones exhibit elevated clonal death without an interface-correlated pattern, despite activation of interface actomyosin and interface JNK signaling. These results suggest receptor-specific downstream pathways of cell elimination, potentially reflecting different combinations of the overlapping mechanisms listed above.

As we can only speculate how these patterns arise, we now include the following section in the discussion.:

Our results reveal distinct apoptotic patterns in tissues containing Toll-receptor-misexpressing clones. These differences likely arise from multiple pro-apoptotic signals acting at and away from the interface. Specifically, our previous work has shown that JNK activation in aberrant and wild type cells at the interface sensitizes both neighboring cells to apoptosis (Prasad, Illek et al. 2023). However, attenuation of JNK signaling only partially suppresses cell death, suggesting the contribution of additional mechanisms. These will certainly include mechanical compression, which can arise from the geometry of interface curvature as well as from population growth dynamics (Bielmeier, Alt et al. 2016, Valon, Matamoro-Vidal et al. 2025). Of note, in a normally developing tissue, cells of different fates meet at lineage boundaries, such as the the anterior-posterior compartment boundary in wing discs. Importantly, these boundaries are generally straight, thereby reducing local neighborhood connectivity that induces JNK in each cell (Prasad, Illek et al. 2023) and eliminating curvature induced compression (Valon, Matamoro-Vidal et al. 2025), which, combined, would strongly reduce interface surveillance mediated apoptosis at these naturally occurring cell fate boundaries. Other effects that influence apoptotic patterns in Toll-receptor-misexpressing clones are buckling of the epithelial sheet in the clone interior (Bielmeier, Alt et al. 2016, Prasad, Illek et al. 2023), mechanical fluctuations at the interface and differential propagation of the resulting tension within cell populations (Schoenit, Monfared et al., 2025), or long-range apoptotic signaling triggered by the elimination process itself (Perez-Garijo, Fuchs et al., 2013). Our boundary-resolved analyses show that Toll-2 and Toll-8 expressing clones generate apoptotic patterns consistent with signals acting at clonal interfaces, whereas wild type encircled Toll-6 and Toll-7 clones exhibit elevated clonal death without an interface-correlated pattern. These results suggest that, despite the common activation of interface actomyosin and JNK signaling, the different long Toll receptors may also engage receptor-specific pathways of cell elimination, and possibly JNK-activation, a view supported by their different functions, ligands and effectors (Akhoyari, Turc et al. 2011, Nakamoto, Moy et al. 2012, Mcllroy, Foldi et al. 2013, Foldi, Anthony et al. 2017, Mishra-Gorur, Li et al. 2019, Lavalou, Mao et al. 2021, Tamada, Shi et al. 2021, Ding, Li et al. 2022, Kong, Zhao et al. 2022, Brutscher and Basler 2025).

3) Similarly, activation of the JNK pathway seems to differ in spatial localisation between experiments with different Tolls. For example, in Toll6 overexpression, TRE-RFP is at the clonal borders on the wildtype side (Fig. 2D'), while in Toll8 overexpression, TRE-RFP response is more diffuse (Fig. 3F). Could the authors comment on this?

Yes, some Toll-like receptors, and especially Toll-6, display unilateral bias. We also observed this for Robo3 and Robo2 to varying degrees. While we work on a mechanistic dissection of these difference, we can only speculate that Toll-2, -6, -7, -8 and other interface surveillance competent molecules described in (Fischer, Ernst et al. 2024) likely engage with different ligands and co-factors in distinct cis- and trans- interactions, creating different inhibitory and activating logic gates at cellular interfaces. The different molecular architectures of cis- and trans-interactions could bias signal readouts unilaterally (preferential activation on one side) or bilaterally (activation on both sides). Transcription factor-driven perturbations that concurrently alter multiple receptors would then tend to sum into bilateral readouts, consistent with our observations in (Fischer, Ernst et al. 2023, Prasad, Illek et al. 2023) and this manuscript. Because of the speculative nature of this discussion, as well as the focus and the word limit of the manuscript, we would like to not extensively comment on it. We hope that our current discussion of different apoptotic patterns, interactions, ligands and molecular effectors conveys enough information about the complexity that underlies these observations.

4) The authors perform a double RNAi of both Toll-2 and Toll-7 and show that this induces JNK activity (Fig. L). What about actin enrichment?

We now include data for both actin enrichment and JNK activation at boundaries of double KD clones.

Reviewer 2: SUMMARY OF THE ADVANCE MADE IN THIS PAPER AND ITS POTENTIAL SIGNIFICANCE TO THE FIELD

In this manuscript, the authors characterize the function of long Toll receptors in interface surveillance, a process protecting from the emergence of cell mutations affecting cell-fate patterning. Toll8 was identified in a genetic screen as a potential mediator of surface surveillance. Since Toll receptors have been identified as regulators of myosin polarity during germ band extension, they reasoned they could modulate actomyosin contractility and adhesion in interface surveillance.

They first analyzed the expression pattern of the Toll family using enhancer trap fluorescent lines. They found specific patterns of Toll-2, Toll-7 and Toll-8, and a ubiquitous expression of Toll-1 in the wing disc. They tested the impact of overexpression of Toll-1, Toll-3 and Toll-4 in clones, but found no sign of a role in interface surveillance such as clones smoothening, JNK activation and apoptosis at the interface. The expression of Toll-6 however (not expressed in the wing disc) induced the formation of a smooth interface with JNK activation and apoptosis at the border. For Toll-8, they found that overexpression induces interface surveillance specifically in region where Toll-8 is not expressed, while Toll-8 RNAi induces it specifically in Toll-8 expression domain, indicating that a discrepancy in Toll-8 level is required. They further tested Toll-2 and Toll-7 for their role in interface surveillance. They found that expression of each of these receptors induces interface surveillance markers, in particular in zone of low endogenous expression. However, their successive inactivation has no effect, while inactivation of both drive interface surveillance in their expression domain, suggesting a redundant function.

They found that Myosin, Rok and Ena accumulate at the interface between different levels of Toll-6 and Toll-8, suggesting that myosin regulation is a core function of Toll receptors.

Then they tested whether aberrant cell fate, known to induce interface surveillance, modifies the expression of Toll receptors. They found that FkhOE and TkvCA impact the expression of Toll-7, Toll-8 and probably Toll-2 in a region-specific way. Ey however only affects Toll-7 and Toll-2.

They further tested the potential implication of Toll receptors at the boundary of tumoral clones and found that Toll 2, 7 and 8 are activated at the border of RasV12 clones. However, their expression pattern is not modified in contexts of cell competition (Myc or Hippo/warts expressing clones).

They tried to impair the discrepancy of Toll 2, 7 or 8 expressions at the border of TkvCA and FkhOE clones using RNAi (for Toll-2, 7 and 8) or overexpression (for Toll-8), but failed to detect any defect in interface surveillance at the border of aberrant cell fate clones. Finally, they showed that Toll-6 and Toll-8 function in interface surveillance does not appear to go through the canonical NF- κ B pathway.

SUGGESTIONS TO AUTHORS

Main concerns:

* My major concern is that although the observations are interesting, convincing and well presented, everything relies on correlation and there is no functional evidence that Toll receptors are required for the interface surveillance process. This weakens the impact of this work. Inactivating Toll-2 and Toll-7 in a context of Ey clones (ideally in a whole domain as done in Fig S7) should be tried since Toll-8 is not affected in this context (although it may be more complex and necessary to inactivate also Toll-6 if Ey leads to an up-regulation of Toll-6).

We fully agree that a direct functional demonstration of necessity remains a key limitation. This issue has been central to our work for several years. Despite several attempts, it has proven impossible to test necessity in this system. We have systematically performed and reported genetic manipulations of cell surface molecules that we identified to be each individually sufficient to induce interface surveillance in (Ey, Fkh or Tkv-) aberrant cells (for example Fig S7.2 in this manuscript, and Fig. S7A-J in (Fischer, Ernst et al. 2024)). However, the cell fate programs induced by Ey, Fkh or Tkv simultaneously regulate many cell surface molecules in overlapping and spatially dynamic patterns within the wing disc. Consequently, removing one or two of them (or bringing them back) cannot abolish the surveillance phenotype, because neighboring cells continue to differ in the expression of other molecules within this recognition code. We also want to emphasize that the experimental challenge here is not just the genetics of these experiments, but also the requirement to precisely reestablish the relative expression levels of multiple surface molecules to match those of surrounding cells. The combinatorial and graded nature of these codes, along with the need to titrate multiple RNAi and UAS transgenes in a single spatial domain, has so far made “necessity” test technically unfeasible.

However, in response to the reviewer’s specific suggestion, we performed a double knockdown of Toll-2 and Toll-7 in Ey-expressing clones (now in Fig S7.1 G,H). Again, we did not observe suppression of interface surveillance in the pouch, consistent with the model that yet additional surface molecules are deregulated in Ey-expressing clones. In fact, Robo2 is ectopically upregulated by Ey in the same domain

- see Fig. 6C in (Fischer, Ernst et al. 2024)) - demonstrating that at least a third surface molecules (out of the 14 molecules we now know of being individually sufficient) is misexpressed in Ey-expressing clones.

To better explain these findings we restructured Fig 6 from the perspective of aberrant cells and expanded our description of the data in the main text. In addition to our conclusions at the end of the result section, we also added this sentence to the discussion section:

While our data demonstrate that individual long Toll receptors are each sufficient to induce interface surveillance, a limitation of our study is the absence of a direct genetic test of their necessity in recognizing aberrant cells, likely due to redundancy arising from the combinatorial deregulation of multiple cell surface molecules.

* How important are Toll receptors at the interface compared to Robo and other genes involved in interface surveillance? How redundant? What do we learn about the interface code? This should be discussed.

Our published and unpublished data show that several receptors, (long Tolls, Robos, Teneurins, and others) are each individually sufficient to elicit interface surveillance. This strongly suggests that the system is not hierarchically organized. We view all molecules as an “equal-opportunity” surveillance network, in which local differences of any individual one trigger the

response. Consistent with this, while we can demonstrate robust sufficiency, our necessity experiments have repeatedly failed. The biological importance of any given cell surface molecule therefore depends primarily on its spatial and temporal expression pattern within which surveillance can occur, rather than an intrinsic hierarchical position in the network. We hope that our restructured Figure 6 and adjustments to the main text and discussion make this point clear.

More specific concerns:

Fig2:

* The quantification of cell death should be in the principal figure, presented the same way (with interface cells distinguished from the rest).

We now include interface-resolved quantification of apoptotic patterns for all long Toll-like receptors (see responses on workflow sensitivity below). The data requires this discussion:

- We previously demonstrated that JNK activation at clonal interfaces sensitizes both mutant and neighboring wild-type cells to apoptosis (Prasad, Illek et al., 2023). Toll-2 and Toll-8 -expressing clones indeed produce apoptotic patterns consistent with JNK activity concentrated at interfaces.
- In contrast, Toll-6 and Toll-7 clones, despite activation of interface actomyosin and interface JNK signaling, exhibit elevated clonal death without an interface-correlated pattern.

The observed apoptotic patterns are almost certainly caused by signals in addition to JNK interface signaling acting at different spatial length scales both near and away from the interface. These additional signals are:

- Mechanical compression at the interface depending on interface curvature, and bulk compression of clones depending on cell growth dynamics (Valon, Matamoro-Vidal et al. 2025).
- Mechanical fluctuations at the clonal interface and differential transduction or propagation of tension within wild type and mutant cell populations (Schoenit, Monfared et al. 2025).
- Long-range apoptotic signals induced by the cell elimination process itself (Perez-Garijo, Fuchs et al. 2013).
- Inhibition of apoptosis by local feed-back mechanisms (Valon, Davidovic et al. 2021).

In addition to these complex signals, differences in apoptotic patterns can also arise from receptor-specific, cell-autonomous activities, potentially determined by distinct ligands, adaptor usage, or structural motifs. Indeed, Toll-6 and Toll-7 have been reported to exhibit tumor-suppressive or pro-survival functions, which would explain relatively lower levels of cell-autonomous JNK and associated apoptosis. Toll-2 and Toll-8, in contrast, promote higher cell autonomous JNK activation and display stronger clone deformation indicating a potential for a stronger cell-autonomous effects on actomyosin regulation than some of the other receptors, which would then impinge on curvature-dependent compression and subsequent apoptosis. (see (Akhouayri, Turc et al. 2011, Nakamoto, Moy et al. 2012, McIlroy, Foldi et al. 2013, Foldi, Anthony et al. 2017, Mishra-Gorur, Li et al. 2019, Lavalou, Mao et al. 2021, Tamada, Shi et al. 2021, Ding, Li et al. 2022, Kong, Zhao et al. 2022, Brutscher and Basler 2025))

Because of the speculative nature of the discussion concerning the precise spatial patterns, we now present quantifications of spatially resolved data in the supplements. Yet to reflect the complexity, we now include this new paragraph in the discussion.:

Our results reveal distinct apoptotic patterns in tissues containing Toll-receptor-misexpressing clones. These differences likely arise from multiple pro-apoptotic signals acting at and away from the interface. Specifically, our previous work has shown that JNK activation in aberrant and

wild type cells at the interface sensitizes both neighboring cells to apoptosis (Prasad, Illek et al. 2023). However, attenuation of JNK signaling only partially suppresses cell death, suggesting the contribution of additional mechanisms. These will certainly include mechanical compression, which can arise from the geometry of interface curvature as well as from population growth dynamics (Bielmeier, Alt et al. 2016, Valon, Matamoro-Vidal et al. 2025). Of note, in a normally developing tissue, cells of different fates meet at lineage boundaries, such as the anterior-posterior compartment boundary in wing discs. Importantly, these boundaries are generally straight, thereby reducing local neighborhood connectivity that induces JNK in each cell (Prasad, Illek et al. 2023) and eliminating curvature induced compression (Valon, Matamoro-Vidal et al. 2025), which, combined, would strongly reduce interface surveillance mediated apoptosis at these naturally occurring cell fate boundaries. Other effects that influence apoptotic patterns in Toll-receptor-misexpressing clones are buckling of the epithelial sheet in the clone interior (Bielmeier, Alt et al. 2016, Prasad, Illek et al. 2023), mechanical fluctuations at the interface and differential propagation of the resulting tension within cell populations (Schoenit, Monfared et al., 2025), or long-range apoptotic signaling triggered by the elimination process itself (Perez-Garijo, Fuchs et al., 2013). Our boundary-resolved analyses show that Toll-2 and Toll-8 expressing clones generate apoptotic patterns consistent with signals acting at clonal interfaces, whereas wild type encircled Toll-6 and Toll-7 clones exhibit elevated clonal death without an interface-correlated pattern. These results suggest that, despite the common activation of interface actomyosin and JNK signaling, the different long Toll receptors may also engage receptor-specific pathways of cell elimination, and possibly JNK-activation, a view supported by their different functions, ligands and effectors (Akhouayri, Turc et al. 2011, Nakamoto, Moy et al. 2012, McIlroy, Foldi et al. 2013, Foldi, Anthony et al. 2017, Mishra-Gorur, Li et al. 2019, Lavalou, Mao et al. 2021, Tamada, Shi et al. 2021, Ding, Li et al. 2022, Kong, Zhao et al. 2022, Brutscher and Basler 2025).

* Please explain how circularity is measured.

We apologize for not including it in the previous manuscript version. We now describe it in the method section.

For Figure 3-4

* A clear explanation is missing on how the different domains (outside/inside Toll expression domain) are determined for circularity, TRE-RFP and Dcp1 intensity quantification.

For TRE-RFP and Dcp1 intensity measurements, we had already provided a detailed description in the *Image Quantification and Statistical Analysis* section and a visual representation of the segmentation logic in Figure S2E. To improve clarity, we have modified Figure S2E and further expanded the text in the *Image Display* section. We hope that this creates the necessary transparency (while allowing us to adhere to the word limits).

Fig3:

* In C and D, TRE-RFP should be shown alone. Also, a staining with Wg would help determine the different domain of the disc (hinge versus pouch, versus notum). Quantifications are missing for Toll8 RNAi. In I and K, the number of n seems quite low for the notum territory.

We now include quantifications for Toll-8 RNAi clones in the revised figure.

Regarding the visualization of TRE-RFP, we agree that JNK patterns are clearer when shown separately. Because the main figure panels are already dense, we have opted to display the corresponding TRE-RFP images in the Supplementary Figures. We retained figures C and D in the main figure to provide an overview of clone morphology and positional context within the disc.

However, to improve orientation within the disc, we have added Phalloidin/F-Actin staining to panels C and D and now also provide the maximum projections of Phalloidin/F-Actin and TRE-RFP side-by-side in the supplements. The Phalloidin/F-Actin signal provides a reference for folds and compartment boundaries, allowing reliable identification of hinge, pouch, and

notum regions.

We now report elevated levels of apoptosis in Toll-8 RNAi clones within the highest Toll-8 expressing regions of the hinge. RNAi clones, optimized for larger size for better visualization distinction of interface JNK, clone shape and actomyosin enrichment, experience reduced apoptosis: due to lower mismatch levels compared to overexpressing clones, lower local activation of JNK along less curved clone interfaces (Prasad, Illek et al. 2023) and reduced curvature-induced compression (Valon, Matamoro- Vidal et al. 2025). Our quantification workflows cannot sensitively resolve this low level of apoptosis, as we detect changes in punctate fluorescence intensity averaged over larger segmented area (inner area, inner band, outer band and outer area), therefore a more subtle increase in apoptotic frequency is diluted by the clone areas (which are optimized to be large). We thus cannot report boundary resolved data for this experiment.

Indeed, the number of n in K is low but were derived from 15 discs. The high Toll-8-expressing region in the notum domain is relatively small; by chance only few clones can be recovered that fall into this region and do not span to regions outside.

Fig4:

*The characterization of cell death pattern should be shown in the principal figure, together with the related quantifications of cell death at the interface of the clone versus further away. In L, the general view of the disc is missing (with TRE-RFP shown independently), together with a zoom on the pouch region, including actin and Dcp1 staining and the corresponding quantifications in the different domains.

We provide quantifications for apoptotic patterns, as discussed in response to Fig 2.

We now include JNK reporter (TRE-RFP) and Actin images for L, as well as quantifications. The double Toll-2; Toll-7 RNAi clones exhibit a markedly lower frequency of apoptotic cells compared to Toll- overexpressing clones, for the same reasons discussed in the response to Fig 3. However, we would be happy to include images, if requested.

Fig6:

* All data presented should be quantified, not only Toll-8 for Fkh and Tkv clones.
 * In the text, the author mentioned: "In our analysis, we found that Fkh and TkvCA-expressing clones exhibited aberrant up and downregulation of Toll-7-Venus and Toll-8-YFP, while Ey-expressing clones altered Toll-7-Venus but not Toll-8-YFP expression (Fig 6)." However, the effect on Toll-8 is not the same for Tkv and Fkh. This should be more carefully commented. Tkv induces downregulation of Toll-8, while Fkh induces an up-regulation. In addition, does Fkh induce an upregulation outside Toll-7 expression domain? This is not commented while clearly shown in the figure.

We now restructured the Figure 6 and S6, incorporated additional spatial quantifications, and revised the text to narrate the data more carefully. The perspective of the figure has been refocused on aberrant cells and the description has been refined to support a conclusion linking three aspects: different transcriptional regulators can modulate the expression of different long Toll receptors in different tissue domains, collectively shaping the surface code. To avoid overcomplicating the description for the interactions of these variables, we have focused the description to align with the new quantifications.

* Why not looking at Toll-6? This is important, independently of the absence of endogenous expression in the wing disc, to explore the idea of the code of interface surveillance.

We completely agree but we do not have a tool for Toll-6 visualization.

* Finally, this figure shows that the code is different depending on the type of cell fate perturbation, which is complex, but interesting. The next step would be to test the necessity of these Toll receptor in interface surveillance. Intents have been made, presented in Fig S7 and S8. However, it would be worth focusing on one particular cell

fate perturbation context such as Ey clones for example. Indeed, since they do not depend on Toll-8, we could expect less redundancy in these clones between different Toll, and be able to test the role of Toll-2, Toll-7 or both.

Please see our detailed response to your first comment.

Fig S1:

* Please explain A and B. There is too little information to appreciate what these two panels bring.

Yes, of course. We now added these legends:

A: Domain structure of the 9 Toll-like receptors in Drosophila (adapted from (Umetsu 2022)). The long Toll-like receptors can be structurally distinguished from the short Toll-like receptors including Toll-1 by their long extracellular domain containing a higher number of Leucine-rich repeats (LRR). Their intracellular domains are also longer and may contain PolyQ-domains.

B: Single Cell RNA-Seq analysis of wild type Drosophila wing imaginal discs as published in (Floc'hlay, Balaji et al. 2023) and accessible online through WingAtlas. Top left panel depicts cell type clusters identified in this data set. Other panels visualize the level of expression for Drosophila Toll-like receptor transcripts Toll-1, Toll-2, Toll-7, Toll-8 and Toll-9. Note that transcription of Toll-3, Toll-4, Toll-5 and Toll-6 was undetectable in the WingAtlas data set.

Fig S6:

* Why using FlyFos-Toll-2 when it does not recapitulate the expression pattern of Toll-2? Why not using Toll-2-GFP used in Fig1 to characterize Toll-2 expression pattern?

We have now done this and replaced all data set with the CRISPRed Toll-2 GFP line. We removed Flyfos data from the manuscript.

* A final recapitulating scheme recapitulating the code of interface surveillance would help.....

Is now provided in Fig 8.

* Precise the number of discs analyzed for each experiment. This information is missing all along

We now included numbers for n and N in all figure legends.

Others:

* Line 457 "our findings highlight the specificity and sensitivity of Toll receptor-mediated detection of aberrant cells..." The sensitivity of the system is an interesting point to discuss, however, the UAS/GAL4 system induces probably huge discrepancies of Tolls. In addition, the fact that loss of function of only Toll2 or Toll7 does not induce interface surveillance, due to redundancy, suggests that a factor 2 in the discrepancy is not sufficient.

We completely agree with the interpretation that UAS/GAL4 and RNAi induce different levels of relative changes, and we indeed attribute the milder phenotype of RNAi-clones at least in part to the lower levels of relative changes. (Yet, we think that redundancy between Toll-2 and Toll-7 is due to functional redundancy.) We have not discussed sensitivity, as we cannot provide a quantitative analysis, or even an approximation, of the magnitude of differences between cells. We are limited by antibody availability to detect the overexpressed proteins, and also feel that to really address this question, we would require a molecularly resolved super-resolution approach quantifying individual molecules. As any discussion would be speculative at this point (and limited in words), we removed the idea of 'sensitivity' from the discussion.

* Fig S8F: it is said that there is no regulation of Dorsal during interface surveillance, however there is an increased signal of Dorsal shown in S8F.

The apparent increase in Dorsal is optical distortion seen when clones create strong basal fold at the interface. We have now included an image of a planar, early-stage clone, supporting our conclusion.

* Line 387: Fig S7M does not exist in this version of the paper. Please correct.

Indeed, this referred to a summary scheme in a previous version of the manuscript. This is now included in Figure 8I.

* 402-404: "Future research needs to address if this is reflected in normal imaginal disc development, such that distinct cell fates establish unique cell surface profiles using a combinatorial code of different long Toll receptors, thereby providing information about the spatial position of cells." This is an interesting point, however, how to explain that normal Toll expression borders in the wing do not lead to Interface surveillance? This should be further discussed.

We now explain this in the discussion:

Importantly, in a normally developing tissue, the graded expression patterns of cell fate established by morphogen gradients would prevent the appearance of strong differences between neighboring cells and thereby prevent the activation of interface surveillance. In contrast, alterations in developmentally aberrant or pre-cancerous cells would activate interface surveillance, thereby correcting mistakes and acting as tumor suppressor mechanism.

And

Of note, in a normally developing tissue, cells of different fates meet at lineage boundaries, such as the the anterior-posterior compartment boundary in wing discs. Importantly, these boundaries are generally straight, thereby reducing local neighborhood connectivity that induces JNK in each cell (Prasad, Illek et al. 2023) and eliminating curvature induced compression (Valon, Matamoro-Vidal et al. 2025), which, combined, would strongly reduce interface surveillance mediated apoptosis at these naturally occurring cell fate boundaries.

* 433-435: "Whether these non-canonical ligand interactions explain our findings that the extracellular domain of a receptor such as Toll-6 is completely sufficient to induce interface surveillance, remains to be investigated." Please simplify and reformulate.

We removed this sentence during restructuring of the discussion, trying to adhere to the word limit.

* Some typos: l. 48 "a distinct tissue-intrinsic surveillance system", l. 52 "consistently induce", l. 71 "all hallmarks", l. 87 "provides" l. 450 "may not to not be" l. 458 "represent", l. 460 "not just in innate immunity".

*Explicit sc-RNAseq (single cell-RNAseq) an AHS (after heat-shock)

Thank you! Corrected.

Some overstatements are found all along the manuscript:

The title "Toll receptors mediate tissue intrinsic surveillance against aberrant cells by detecting cell fate aberrations" does not represent the actual data in the present version of the manuscript since no clear proof of the role of Toll receptor in interface surveillance is established.

We completely understand your concerns and agree that the lack of necessity evidence is a limitation of our study. Yet, we define tissue-intrinsic surveillance by the entire repertoire of observed responses (JNK, actomyosin and apoptosis) and show that each long Toll receptor is sufficient to induce it. We like the current title because it is succinct and a similar title was approved for our previous manuscript, despite the same experimental limitations (Fischer, Ernst et al. 2024).

Line 29: "key mediators"

We removed 'key'.

Line 383: "our study demonstrates a novel function of long Toll receptors..." : This is not demonstrated since no loss of function experiments are conclusive on this point.

We now state:

Our study demonstrates that long Toll receptors are each individually sufficient to induce interface surveillance via detection of expression level differences between neighboring cells, phenocopying the responses induced by the presence of cells with aberrant cell fate programs.

Line 405: "Our findings emphasize the importance of a spatial code of Toll receptor for the detection of aberrant cell fates..." This is an overstatement. These findings "suggest" the importance of a spatial code.

Absolutely, corrected.

REFERENCES

Akhouayri, I., C. Turc, J. Royet and B. Charroux (2011). "Toll-8/Tollo negatively regulates antimicrobial response in the *Drosophila* respiratory epithelium." *PLoS Pathog* **7**(10): e1002319.

Bielmeier, C., S. Alt, V. Weichselberger, M. LaFortezza, H. Harz, F. Julicher, G. Salbreux and A. K. Classen (2016). "Interface Contractility between Differently Fated Cells Drives Cell Elimination and Cyst Formation." *Curr Biol* **26**(5): 563-574.

Brutscher, F. and K. Basler (2025). "Functions of *Drosophila* Toll/NF-kappaB signaling in imaginal tissue homeostasis and cancer." *Front Cell Dev Biol* **13**: 1559753.

Ding, X., Z. Li, G. Lin, W. Li and L. Xue (2022). "Toll-7 promotes tumour growth and invasion in *Drosophila*." *Cell Prolif* **55**(2): e13188.

Fischer, F., L. Ernst, A. Frey, K. Holstein, D. Prasad, V. Weichselberger, R. Balaji and A. -K. Classen (2023). "A cell surface code mediates tissue-intrinsic defense against aberrant cells in epithelia." *bioRxiv*: 2023.2002.2016.528665.

Fischer, F., L. Ernst, A. Frey, K. Holstein, D. Prasad, V. Weichselberger, R. Balaji and A. K. Classen (2024). "A mismatch in the expression of cell surface molecules induces tissue- intrinsic defense against aberrant cells." *Curr Biol* **34**(5): 980-996 e986.

Floc'hlay, S., R. Balaji, D. Stankovic, V. M. Christiaens, C. Bravo Gonzalez-Blas, S. De Winter, G. J. Hulselmans, M. De Waegeneer, X. Quan, D. Koldere, M. Atkins, G. Halder, M. Uhlirva, A. K. Classen and S. Aerts (2023). "Shared enhancer gene regulatory networks between wound and oncogenic programs." *Elife* **12**.

Foldi, I., N. Anthoney, N. Harrison, M. Gangloff, B. Verstak, M. P. Nallasivan, S. AlAhmed, B. Zhu, M. Phizacklea, M. Losada-Perez, M. Moreira, N. J. Gay and A. Hidalgo (2017). "Three-tier regulation of cell number plasticity by neurotrophins and Tolls in *Drosophila*." *J Cell Biol* **216**(5): 1421-1438.

Kong, D., S. Zhao, W. Xu, J. Dong and X. Ma (2022). "Fat body-derived Spz5 remotely facilitates tumor-suppressive cell competition through Toll-6-alpha-Spectrin axis-mediated Hippo activation." *Cell Rep* **39**(12): 110980.

Lavalou, J., Q. Mao, S. Harmansa, S. Kerridge, A. C. Lellouch, J. M. Philippe, S. Audebert, L. Camoin and T. Lecuit (2021). "Formation of polarized contractile interfaces by self-organized Toll-8/Cirl GPCR asymmetry." *Dev Cell* **56**(11): 1574-1588 e1577.

McIlroy, G., I. Foldi, J. Aurikko, J. S. Wentzell, M. A. Lim, J. C. Fenton, N. J. Gay and A. Hidalgo (2013). "Toll-6 and Toll-7 function as neurotrophin receptors in the *Drosophila melanogaster* CNS." Nat Neurosci **16**(9): 1248-1256.

Mishra-Gorur, K., D. Li, X. Ma, Y. Yarman, L. Xue and T. Xu (2019). "Spz/Toll-6 signal guides organotropic metastasis in *Drosophila*." Dis Model Mech **12**(10).

Nakamoto, M., R. H. Moy, J. Xu, S. Bambina, A. Yasunaga, S. S. Shelly, B. Gold and S. Cherry (2012). "Virus recognition by Toll-7 activates antiviral autophagy in *Drosophila*." Immunity **36**(4): 658-667.

Perez-Garijo, A., Y. Fuchs and H. Steller (2013). "Apoptotic cells can induce non-autonomous apoptosis through the TNF pathway." Elife **2**: e01004.

Prasad, D., K. Illek, F. Fischer, K. Holstein and A. K. Classen (2023). "Bilateral JNK activation is a hallmark of interface surveillance and promotes elimination of aberrant cells." Elife **12**.

Schoenit, A., S. Monfared, L. Anger, C. Rosse, V. Venkatesh, L. Balasubramaniam, E. Marangoni, P. Chavrier, R. M. Mege, A. Doostmohammadi and B. Ladoux (2025). "Force transmission is a master regulator of mechanical cell competition." Nat Mater **24**(6): 966-976.

Tamada, M., J. Shi, K. S. Bourdot, S. Supriyatno, K. H. Palmquist, O. L. Gutierrez-Ruiz and J. A. Zallen (2021). "Toll receptors remodel epithelia by directing planar-polarized Src and PI3K activity." Dev Cell **56**(11): 1589-1602 e1589.

Umetsu, D. (2022). "Cell mechanics and cell-cell recognition controls by Toll-like receptors in tissue morphogenesis and homeostasis." Fly (Austin) **16**(1): 233-247.

Valon, L., A. Davidovic, F. Levillayer, A. Villars, M. Chouly, F. Cerqueira-Campos and R. Levayer (2021). "Robustness of epithelial sealing is an emerging property of local ERK feedback driven by cell elimination." Dev Cell **56**(12): 1700-1711 e1708.

Valon, L., A. Matamoro-Vidal, A. Villars and R. Levayer (2025). "Interfacial tension and growth both contribute to mechanical cell competition." Curr Biol **35**(21): 5372-5383 e5374.

Second decision letter

MS ID#: dev.205006R1

MS TITLE: Toll receptors mediate tissue intrinsic surveillance against aberrant cells by detecting cell fate aberrations

AUTHORS: Anna Frey, Laurin Ernst, Friedericke Fischer, Lale Alpar, Yohanns Bellaiche and Anne-Kathrin Classen

Dear Anne-Kathrin,

I have now received all the referees reports on the above manuscript, and have reached a decision. The referees' comments are appended below.

The overall evaluation is very positive and we would like to publish your manuscript in Development. In order to proceed with formal acceptance could you please address the minor issues mentioned by there reviewer.

Reviewer 1

Advance summary and potential significance to field

The authors have done an outstanding job at addressing my comments and those of the other reviewer. They have improved further the quality and readability of their study, which was already high. The added discussion about apoptosis is excellent.

Here are a few minor things I have noticed for correction:

- Line 60: typo in "Additonally"
- Lines 85-89: in these sentences about the work linking Toll receptors to a "cell surface code" in *Drosophila* embryos, please include Lavalou et al 2021 and Sharrock et al 2022. These studies are cited later, but they also need to be cited here in this context.
- 1130 "Toll-2-LacZ transcriptional reporter (H), or a FlyFos-Toll-2-GFP construct": presumably the mention of FlyFos-Toll-2-GFP needs to be removed.

Second revisionAuthor response to reviewers' comments

Reviewer 1: The authors have done an outstanding job at addressing my comments and those of the other reviewer. They have improved further the quality and readability of their study, which was already high. The added discussion about apoptosis is excellent.

Here are a few minor things I have noticed for correction:

- Line 60: typo in "Additonally"
 - Corrected!
- Lines 85-89: in these sentences about the work linking Toll receptors to a "cell surface code" in *Drosophila* embryos, please include Lavalou et al 2021 and Sharrock et al 2022. These studies are cited later, but they also need to be cited here in this context.
 - Corrected!
- 1130 "Toll-2-LacZ transcriptional reporter (H), or a FlyFos-Toll-2-GFP construct": presumably the mention of FlyFos-Toll-2-GFP needs to be removed.
 - Corrected! (This was actually already corrected in the Supplemental Data file.)

Third decision letter

MS ID#: dev.205006R2

MS TITLE: Toll receptors mediate tissue intrinsic surveillance against aberrant cells by detecting cell fate aberrations

AUTHORS: Anna Frey, Laurin Ernst, Friedericke Fischer, Lale Alpar, Yohanns Bellaiche and Anne-Kathrin Classen

Dear Anne-Kathrin,

I am happy to tell you that your manuscript has been accepted for publication in *Development*, pending our standard publication integrity checks.